# Valuing Ecosystem Services Provided by Pasture-Based Beef Farms in Alentejo, Portugal

**Manuel P. dos Santos *** [ID]**, Tiago G. Morais** [ID]**, Tiago Domingos** [ID] **and Ricardo F. M. Teixeira** [ID]

MARETEC—Marine, Environment and Technology Centre, LARSyS, Instituto Superior Técnico, Universidade de Lisboa, Av. Rovisco Pais, 1, 1049-001 Lisbon, Portugal
* Correspondence: manueldossantos@tecnico.ulisboa.pt

**Abstract:** This work aims to measure and value the ecosystem services of grasslands and croplands covered by pasture-based beef farms in Alentejo. It combines pixel-level data from the Portuguese Mapping and Assessment of Ecosystem Services study and farm-level data from 40 farms. Five ecosystem services were considered: soil protection, carbon sequestration, support to extensive animal production, plant food production and fiber production. Two different approaches for service quantification were used: an "average class" method and a "buffer" approach. Double counting issues were avoided by applying a specific methodology developed for this study. The results obtained were similar for both approaches in the case of grasslands, with an average value between 146 and 176 €/ha/year. For croplands, the average service value oscillated between 40 and 166 €/ha/year. Soil protection was the most valuable service, with over 90% of the total value. Extrapolating these results for the entire region, the five ecosystem services were estimated to be worth between 173 M€ (class method) and 223 M€ (buffer approach). These results suggest that pasture-based beef farms in Alentejo help to provide a significant number of ecosystem services with positive environmental effects that are currently not remunerated by the market.

**Keywords:** economic valuation; ecosystem services; grasslands; beef farms; pasture; agriculture; environment; Alentejo; *montado*; *dehesa*

## 1. Introduction

Livestock provides a relevant fraction of the protein that humans consume [1]. Within the livestock sector, beef farming has gained a reputation as one of the most polluting food production systems [2]. However, beef production includes a vast range of production systems and subsystems; a proper quantitative impact evaluation should take into account, as far as possible, local conditions and mitigating factors [3–5]. The absence of such an analysis, at local and global scales, gives rise to the risk of acting on general observations and heuristics, which may cause unwanted results. Environmentally friendly local/regional systems may get categorized as unsustainable simply due to the product they produce. It is especially important to avoid these kinds of perverse effects in areas where livestock is key for managing and enhancing ecosystems and their services [6].

Ecosystem services were first defined as all of the benefits that human populations derive directly or indirectly from ecosystem functions [7]. This concept has developed to include certain hidden contributions of ecosystems and gained momentum after the Millennium Ecosystem Assessment [8]. However, along with services, ecosystems also provide dis-services that affect well-being by reducing productivity and increasing production costs [9]. The net ecosystem services, i.e., positive services minus negative services, affect human well-being. The concept is especially relevant in agroecosystems that are important providers of ES, in the sense that they provide a more diversified set of economic, environmental, cultural and social goods and services [10]. The ecosystem services (ES) approach has a great potential to link ecosystem conservation and the sustainable use of

natural resources; however, due to limited funding and resources, the concept has not been widely implemented [11].

One attempt to measure ES and link them with sustainability issues in Portugal was the Portuguese Mapping and Assessment of ES (PT-MAES) [12]. This work was a pilot study within the framework of the European Union Biodiversity Strategy of 2020, a pan European initiative launched by the European Commission. The PT-MAES report and results focus on the Alentejo region. Alentejo, located in the south of Portugal, is the largest NUTS II region in the country, accounting for 54% of national utilized agricultural area [13]. The Mediterranean climate, soils and topographic characteristics of the region favor extensive beef cattle production, which is one of the main agricultural outputs from the region [14]. A significant fraction of the territory used for cattle production is part of an agroforestry system called *dehesa* in Spain and *montado* in Portugal. This is a mixed agricultural/pasture ecosystem coexisting with medium/low densities of trees (cork/holm oak and oak-based). It has been shown to be very well adapted to livestock production and ensures diversification of farmer income through the supply of additional forestry outputs like cork while enhancing the provision of ecosystem services [15]. Alentejo is therefore a prime example of a region where animal management may be key to the health and management of ecosystems.

In this work, we perform economic and environmental assessments of the effects of pasture-based beef farms (PBBF) in Alentejo as providers of ES. The values from PT-MAES were combined with real farm data to account for ES supply level. The ES considered were soil protection (SP), in terms of avoided erosion, carbon sequestration (CS), support to extensive animal production (SEAP), plant food production (PFP) and fiber production (FP). Two different methods were used to determine the ES level of the PBBF of the dataset. In the first approach, for each ES, we applied the average regional service class level. In the second approach, we estimated the average value of each service level, considering a buffer around the geographic coordinates of each farm. A regional extrapolation was then performed, departing from a farm-level assessment to estimate the total ES value promoted by PBBF in the Alentejo region.

## 2. Materials and Methods

### 2.1. Characterization of the Study Region

#### 2.1.1. Study Region

The Alentejo region makes up a significant fraction of the south of Portugal. Of the approximately 2 million ha of useful agricultural area in Alentejo, 67% comprises permanent pastures. There are approximately 1.4 million bovines in mainland Portugal, 56% of which are in Alentejo [16]. The typical pasture-based beef production system consists of raising the calves on the farm. On average, pasture-based beef farms in Alentejo have 182 ha and 98 livestock units (LU) and occupy 0.39 full-time equivalent annual work units [17]. There are additional specialized fattening farms in the region, based on grazing plus roughage or concentrate feed regimes until animals reach the required weight for slaughter [18].

#### 2.1.2. Farm Data

Farm data were collected in the context of the Animal Future project (SusAn/0001/2016). The purpose of this project was to study ways to increase the sustainability of animal production systems in Europe. This was mainly achieved by three actions: measuring the different aspects of sustainability in animal farming to make an inventory of innovative practices applied in different European regions, evaluating the impacts of innovations on sustainability, and devising strategies to promote the adoption of innovation practices. Data from farms were collected by the authors between May and October of 2019 through personal interviews with farmers. In total, 40 farms were sampled in this work, whose location can be observed in Figure 1, where the names of the farms were omitted due to privacy issues. Contact with farmers was established mainly through former collaborations with scientific projects (47%) and by producer association referencing (39%). About 35% of

the surveyed farms have land within the Natura 2000 Network and 32% produce according to organic production standards. Each farmer was interviewed individually using the survey in Appendix A, which included 119 questions that address general farm information (17 questions) and environmental (46), economic (28) and social (28) dimensions. Examples of general questions were "area of the farm", "legal form" and "years of experience". Specific environmental questions were related to, among other things, indicators of herd characteristics, fertilizers used and energy consumption. The economic component mainly regarded outputs and costs. The social dimension comprised questions such as hours worked per week, work-life balance or succession in the farm. From the total number of sampled farms, 31 answered the social component completely while the other 13 farmers preferred not to do so. In the environmental and economic compartments, in the absence of information or refusal from the farmer to answer, regional and/or representative defaults were applied.

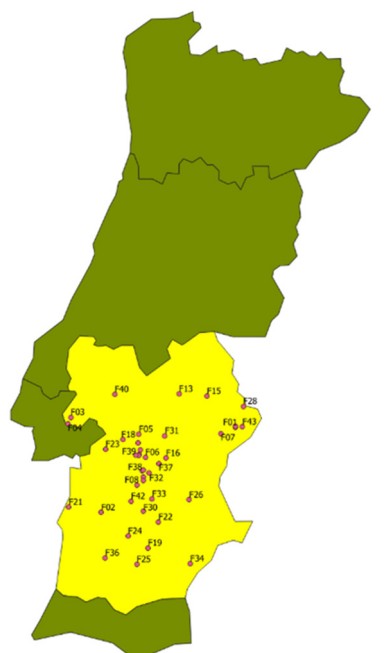

**Figure 1.** Location of surveyed farms within the Alentejo region (NUTS II), Portugal.

*2.2. Portuguese Mapping and Assessment of Ecosystem Services (PT-MAES)*

In this study, the ecosystem service (ES) levels were obtained from a PT-MAES assessment considering five ES services. Soil protection (SP) level was estimated through the contribution to reducing soil erosion by comparing it with the worst-case scenario (i.e., land cover that would generate the highest erosion rate at a given point). Soil erosion rates were estimated using the Universal Soil Loss Equation.

Carbon sequestration (CS) was estimated as the balance of gains and losses of carbon in biomass (above and belowground), considering the land use transitions that occurred between 1990 and 2007 (assuming these transitions occurred at a constant rate). Emission/sequestration coefficients were obtained from the Portuguese National Inventory Report (NIR) for greenhouse gases emissions for each land use transition [19].

Support to Extensive Animal Production (SEAP) was quantified and mapped by determining average livestock densities (for calves, dairy cattle and sheep) in pasture areas within the study region, using official national statistics at the Municipality level. As it was impossible to geographically identify pastures where each type of animal production occurs, average livestock density was estimated considering the two main species together (cows and sheep) for the given pasture area in the national statistics for each municipality.

Plant food production (PFP) was assessed based on the establishment of a correspondence between the main crops in the study area with the European Nature Information

System (EUNIS) ecosystems. Once the correspondence with land use classes had been made, the average productivity was estimated for each ecosystem.

Fiber production (FP) was estimated through the mean annual increments of forest trees, as presented in the Portuguese National Inventory Report, deducting biomass losses due to natural mortality.

### *2.3. Modelling Approaches*

To evaluate the ES levels at a farm level and at the entire dataset level, two different approaches were applied. The "class approach" assumed that the surveyed farms were representative of the PBBF of Alentejo and used the average ES level of the entire region for each relevant ecosystem present in the sampled farms, namely grasslands, *dehesa* and croplands. The "buffer approach" used the coordinates of each farm and generated a buffer with a radius of 1 km. From that buffered area, an average ES level for each relevant ecosystem in the farm was computed. A more detailed explanation of each approach is available in the following sections. The two approaches were then subjected to economic valuation and a comparison of results.

#### 2.3.1. Modelling Approach by Average Class (AC)

In PT-MAES, the studied ecosystems are named according to the EUNIS classification system [20]. Only the relevant areas in the context of PBBF, pastures and croplands, were analyzed under this approach. To match these areas from sampled farms with the EUNIS classification, a classification key was produced according to the characteristics of each ecosystem. According to this classification, all pasture areas (natural and sown) were assumed to correspond to the EUNIS class "dry grasslands or *dehesa*" and all croplands (for animal consumption or not) to "arable land and market gardens".

For each ES, PT-MAES presents a scale of service level by ecosystem. This scale was divided by classes, with a correspondent fraction of area that presents each level of service. We computed an average for the overall service level by multiplying each fraction of area with the mid-point value of its class or, in its absence, with the upper value of the service level. The proportion between grassland and *dehesa* presented in PT-MAES was taken into account when computing the average values for the pasture areas of the surveyed farms.

#### 2.3.2. Modelling Approach by Buffer Generation (BG)

In this approach, we combined the location of each farm with PT-MAES data to estimate the level of each ES. The geographic coordinates of each farm were obtained directly during the visit to the farm and/or the interviews. Since real farm limits were unavailable, we generated a buffer with a radius of 1 km around the geographic coordinates of each farm headquarters. We computed the average ES level from this area for each relevant ecosystem.

Data for ES were based on a map with a scale of 1:100,000. To calculate the average ES level in the 1 km buffer area of each farm, a geographic information system software was used (QGIS software version 3.16.2, Open Source Geospatial Foundation Project, Chicago, IL, USA. http://qgis.osgeo.org (accessed on 10 November 2022)).

### *2.4. Ecosystem Services Economic Valuation*

For each ES, different data sources were used to convert environmental benefit/damage into an economic value. For SP, the economic value was 5.03 € per ton of avoided erosion [21]. This value includes sediment and nutrient losses and is a previous replacement cost estimation. Since the reference value was for the year 2007 (SP2007), it was updated to 2020 price levels (SP2020) using the yearly inflation rate between 2007 and 2019 (IFi):

$$\mathrm{SP}_{2020}\left(\frac{\text{€}}{\mathrm{t}}\text{of avoided erosion}\right) = \mathrm{SP}_{2007} \times \prod_{i=2007}^{2020}(1 + \mathrm{IF_i}). \tag{1}$$

For CS valuation, the most common practice is to apply the social cost of carbon. This term represents the economic cost caused by an additional ton of carbon dioxide emissions or its equivalent and varies according to the applied discount rate, productivity growth and temperature sensitivity [22]. There is no academic consensus about this value [23], as it depends on many variables, assumptions and on the context. Other options are to use the carbon market price or the shadow price. In Table 1, we present the list of values applied for the CS valuation [24].

**Table 1.** Carbon price estimates based on different explicit carbon pricing mechanisms [24].

| Pricing Mechanism | Source | Details | Value €/tCO$_2$eq |
|---|---|---|---|
| Market Price | Daily Carbon Prices (ember-climate.org) | European market quotation for June 2021 (average) | 53 |
| Social Cost | [25] | Estimates for Social Cost of Carbon | 37.5 |
| Shadow Price | [26] | Upper bound value for the 2030 range estimate | 92 |

For the estimation of the value of support to extensive animal production (SEAP), we used data from regional agricultural accountancy statistics for beef farms [13]. We started by calculating the average rent value per hectare without subsidies or taxes, i.e., 14 €/ha/year. The average added value per ha includes three sources of added value: animal production, vegetal production and other production. Here, we considered the proportion related to animal production (67%) and applied it to the calculated rent value per ha, resulting in 8.72 €/ha of rent due to animal production. As the SEAP service level is measured in terms of LU, we applied the regional value of 0.5 LU/ha to stipulate an overall price of 17.43 €/LU for this ES.

To estimate the value of plant food production, a similar method to that explained for SEAP was applied. Departing from the same rent value, 14 €/ha, here, we used the proportion of added value generated by vegetal production (14%). We estimated a value of 1.89 €/ha of rent due to vegetal production. Considering the average productivity of the relevant crop basket for the region, i.e., 4.91 t/ha [16], we estimated a price of 0.39 € per ton for the valuation of this ES.

For fiber production, we considered the volume of the main three species used for fiber production in the region: eucalyptus, maritime pine and stone pine [27]. We used the same source to infer the regional proportion of each species in terms of biomass volume. We then multiplied the proportion of each species by the respective price (based on the terrain prices). This calculation delivered a regional price of 21.5 € per m3 for the valuation of fiber production.

To perform the valuation, the estimated ES levels (ESlevel) were multiplied by the values explained in the previous section (ESprice) to calculate the ESvalue of each ES, as follows:

$$\text{ES}_{\text{value}}\left(\frac{\text{€}}{\text{ha.year}}\right) = \text{ES}_{\text{level}}\left(\frac{\text{i}}{\text{ha.year}}\right) \times \text{ES}_{\text{price}}\left(\frac{\text{€}}{\text{i}}\right), \tag{2}$$

where i represents the relevant unit for each ES.

As mentioned in Section 2.3.1, the farm-level dataset included information about agricultural occupations and respective areas, which were organized into four main groups: natural pastures, sown pastures, crops and other cultures. The first two were assumed to correspond to ecosystem grasslands or *dehesa* (Ag&d), while the other two corresponded to croplands (Acrop). Summing the ESvalue applicable for each ES and ecosystem, we obtained the Total Ecosystem Value (TEV) per ecosystem. It is also possible to estimate the total value of the ES provided by each farm (ESfarm), as follows:

$$\text{ES}_{\text{farm}}\left(\frac{\text{€}}{\text{year}}\right) = \text{TEV}_{\text{g\&d}}\left(\frac{\text{€}}{\text{ha.year}}\right) \times \text{A}_{\text{g\&d}}(\text{ha}) + \text{TEV}_{\text{crop}}\left(\frac{\text{€}}{\text{ha.year}}\right) \times \text{A}_{\text{crop}}(\text{ha}). \tag{3}$$

For the modelling approach of a buffer around farm location, the ES values provided yearly by each farm ($ES_{farm}$) were obtained directly by the multiplication of the valuation per unit (i) by the average service level of the buffered area:

$$ES_{farm}\left(\frac{\text{€}}{\text{year}}\right) = ES_{price}\left(\frac{\text{€}}{\text{i}}\right) \times ES_{level}\left(\frac{\text{i}}{\text{ha.year}}\right) \times A_{farm}(\text{ha}). \tag{4}$$

With the previously estimated values, we performed an extrapolation of the ES value provided by PBBF for the entire Alentejo region. Multiplying the area of permanent pastures in the region reported by national statistics by the calculated TEV (€/ha/year), we estimated the total value of ES provided by PBBF in Alentejo (assuming that all permanent pastures are used by PBBF). Moreover, assuming the same proportion of grasslands and *dehesa* vs. cropland area verified in the farm-level dataset applied to the region, it was possible to estimate the value of ES provided by the area of croplands associated with PBBF by summing the two components:

$$\text{Value of ES on Alentejo}\left(\frac{\text{€}}{\text{year}}\right) = TEV_{g\&d}\left(\frac{\text{€}}{\text{ha.year}}\right) \times A_{g\&d\_Alentejo}(\text{ha}) + TEV_{crop}\left(\frac{\text{€}}{\text{ha.year}}\right) \times A_{crop\_PBBF\_Alentejo}(\text{ha}). \tag{5}$$

## 3. Results

This section is divided by subheadings. It is intended to provide a concise and precise description of the experimental results, their interpretation, as well as the experimental conclusions that can be drawn.

### 3.1. Characterization of the Sampled Farms

Figure 2 presents the sampled area in terms of land use class, ecosystem type and total area per farm. The sampled farms are heterogenous in terms of total area but also in terms of the land use classes. The median area of grasslands is about 602 ha per farm, ranging between about 23 ha (Farm 16) and 3450 ha (Farm 03). Among the sampled farms, 35% (14 farms) only have natural pastures and only 2 have sown pastures, while all others have both pasture systems (60%—24 farms). Farm 03 has the highest area of grasslands, i.e., 3450 ha of sown pasture (this farm does not have natural pasture). Farm 16 only uses natural pastures, and it has also the lowest grassland area among the sampled farms. There are 9 farms with grassland only. The median area of croplands is 33 ha, ranging between zero and 300 ha (Farm 22). Among the farms with cropland, about 87% (16 farms) have crops for animal consumption, and only 40% (13 farms) have other plant production.

The land use class with the largest variation (interquartile distance) in area between farms is the natural pastures class, i.e., about 270 ha, while the class with the lowest variation is the other plant production class, with only 10 ha. The interquartile distances of sown pastures and crops for animal consumption are 165 ha and 61 ha, respectively. Due to the higher variation of both pasture systems in comparison with the variation of crops production classes, the aggregated land use classes of grassland have a significantly higher variation than of croplands, with interquartile distances of 288 ha and 67 ha, respectively.

Regarding the complete set of sampled farms, grasslands and *dehesa* classes account for 91% of the total area of the dataset (about 22,370 ha). Cropland accounts for 9% (about 2232 ha). Within grasslands, natural pastures have a higher proportion than sown pastures, i.e., 66% (14,757 ha) and 34% (7613 ha), respectively. In croplands, crops for animal consumption represent 86% of the area (1998 ha), while other productions represent only 14% (234 ha).

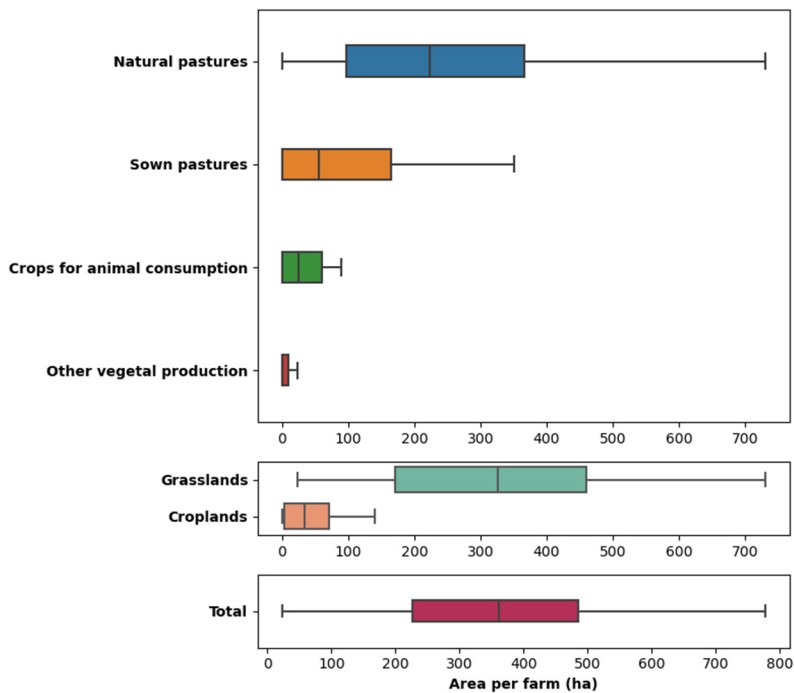

**Figure 2.** Areas in the sample for each land use class, ecosystem type (grassland and cropland), and total area per farm. "Grassland" area is the sum of "Natural pastures" and "Sown pastures", and "Croplands" is the sum of "Crops for animal consumption" and "Other vegetal production".

*3.2. Ecosystem Services Valuation*

Table 2 presents the average ES level per unit of area for the two modelling approaches. In general, the buffer approach leads to higher values than the class approach. The only exception is for SEAP in the grassland and *dehesa* land use ($-0.1$ LU/ha). The most significant difference occurs in the SP service. For the grassland and *dehesa* land use, the difference is 20 t/ha/year (buffer approach: 34.5 t/ha/year; class approach: 24.5 t/ha/year). For croplands, the difference is 6.4 t/ha/year (buffer approach: 34.5 t/ha/year; class approach: 28.1 t/ha/year). The difference between modelling approach for other ES is not significant. For example, excluding SP, the highest difference is only 1.8 t/ha in PFP in croplands. The two modelling approaches also lead to equal ES levels for 3 out 5 ES for grasslands (CS, PFP and FP) and 1 out of 5 for croplands, i.e., SEAP.

**Table 2.** Average ecosystem service level per unit of area for the two modelling approaches.

| Ecosystem Service | Units | Approach by Average Class | | Approach by Buffer Generation | |
| --- | --- | --- | --- | --- | --- |
| | | Grasslands and *Dehesa* | Croplands | Grasslands and *Dehesa* | Croplands |
| Soil protection (SP) | t/ha/year | 28.1 | 14.5 | 34.5 | 34.5 |
| Carbon sequestration (CS) | t C/ha/year | $-0.2$ | $-0.6$ | $-0.2$ | $-0.3$ |
| Support to extensive animal production (SEAP) | LU/ha | 0.4 | 0.0 | 0.3 | 0 |
| Plant food production (PFP) | t/ha | 0.0 | 1.5 | 0 | 3.3 |
| Fiber production (FP) | m$^3$/ha | 0.4 | 0.0 | 0.4 | 0.2 |

Analysing the results for PBBF per ha (Table 3), the average of the total ES valuation (sum of all ES) is different for the two modelling approaches. The valuation is nearly 34% higher with the buffer generation approach (131 €/ha vs. 175 €/ha). Observing each ES separately, SP dominates the total ES valuation for both approaches: 131.7 €/ha using the

class approach and 173.4 €/ha using buffers. CS is the only ES with a negative average valuation in both modelling approaches (class: −13.7 €/ha; buffer: −12.2 €/ha, applying the market price). The average value for SEAP in the two modelling approaches is similar between approaches, but the minimum estimated value differs significatively (class: 2.9 €/ha; buffer: 0 €/ha). PFP present very low values under both approaches, i.e., between 0 €/ha an 0.03 €/ha in the approach by class and between 0 €/ha and 1.3 €/ha in the second method. For FP, both approaches present similar average values (class: 7.4 €/ha; buffer: 8.3.6 €/ha), but the buffer approach presents a much higher variation, ranging between 0 €/ha and 53.1 €/ha, while under the class approach, it ranges between 4.1 €/ha and 8.5 €/ha).

**Table 3.** Ecosystem services valuation in €/ha for the PBBF of the dataset.

| Ecosystem Service | Average Value Per ha | | Minimum Value Per ha | | Maximum Value Per ha | |
|---|---|---|---|---|---|---|
| | AC | BG | AC | BG | AC | BG |
| SP | 131.7 € | 173.4 € | 104.1 € | 18.4 € | 141.3 € | 1153.9 € |
| CS | −13.7 € | −12.2 € | −22.6 € | −54.2 € | −10.5 € | 86.8 € |
| SEAP | 5.5 € | 5.1 € | 2.9 € | 0.0 € | 6.4 € | 8.8 € |
| PFP | 0.1 € | 0.2 € | 0.0 € | 0.0 € | 0.3 € | 1.3 € |
| FP | 7.4 € | 8.3 € | 4.1 € | 0.0 € | 8.5 € | 53.1 € |
| Total | 131.0 € | 174.9 € | 88.8 € | 16.4 € | 145.8 € | 1166.1 € |

SP—Soil protection; CS—Carbon sequestration; SEAP—Support to extensive animal production; VFP—Vegetal food production; FP—Fiber production; AC—Approach by average class; BG—Buffer generation.

Figure 3 presents the average, minimum and maximum valuations per ES at the farm level, as well as the totals resulting from the sum of the individual valuations of all ES within each farm. SP is clearly the ES that contributes the most to ES valuation at the farm level, i.e., 99% for the class approach and 102% for the buffer approach (over 100% due to the negative contribution of CS). CS presents a negative value for all farms under the approach by class, but under the buffer approach, 10 out of the 40 farms present a positive value up to a maximum of 8794 € for a single farm. SEAP has a similar proportion in both approaches (4% for both). On average, PFP only contributes 0.04% and 0.08% to the total valuation at the farm level for the class and buffer approaches, respectively. The minimum value for PFP is 0 €/farm for both approaches, but the maximum per farm is more than double that of the buffer approach (170 €/ha vs. 460 €/ha). The inverse situation occurs with FP, representing 6% of the total ES value in the class approach compared to only 4% in the buffer approach. Despite these differences, the average total value of the ES provided per farm is similar for both methods, i.e., 81,719 € for the class approach and 86,663 € with the buffer approach. The total value of ES provided by a single PBBF ranges between 3352 € and 502,923 € in the first method and between 928 € and 851,271€ in the second.

Table 4 presents the sum of ES valuations for the entire dataset considered in this work. SP is by far the most valuable ES provided by PBBF, accounting for almost the total value in both methods. SEAP represents 4% of the ES value in both approaches, being the second highest value with the buffer approach but the third using the class approach, in which FP presents the second highest value. PFP represents a very low value, nearly 0%, for both approaches. CS has a negative effect in both methods, from 6% to 18% of the total dataset ES value, depending on the considered carbon valuation. The total ES value attributable to the PBBF of the dataset adds up to 3,205,429 € applying the class approach and 3,466,401 € using buffers (applying carbon market prices).

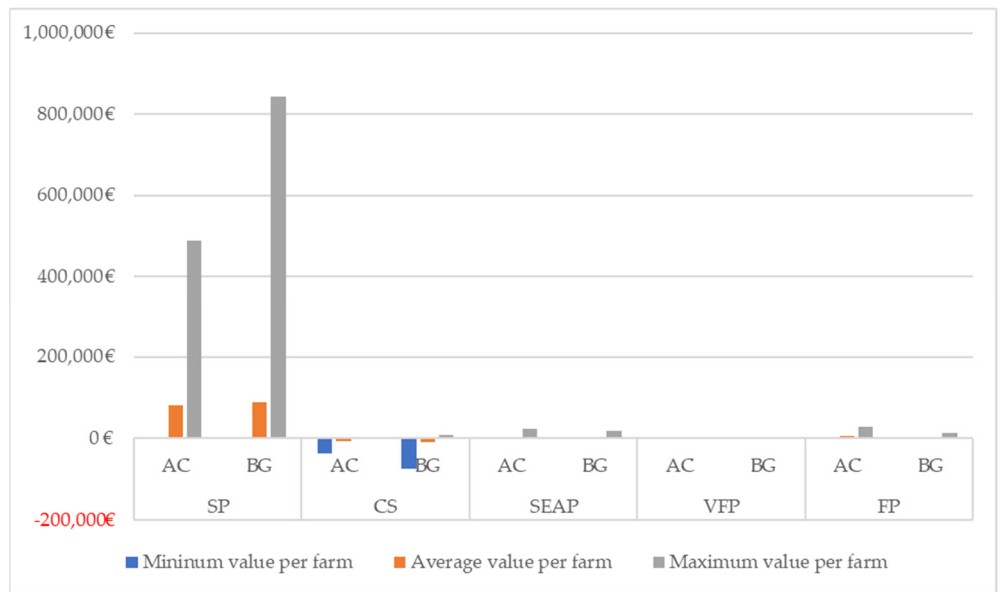

**Figure 3.** Graphical comparison of the economic values of each ecosystem service per farm. SP—Soil protection; CS—Carbon sequestration; SEAP—Support to extensive animal production; VFP—Vegetal food production; FP—Fiber production; AC—Approach by average class; BG—Buffer generation.

**Table 4.** Total ecosystem services value for all farms in the dataset on aggregate, using the class (AC) and buffer (BG) approaches.

| Ecosystem Service | Approach by Average Class (AC) | | Approach by Buffer Generation (BG) | |
|---|---|---|---|---|
| | AC | % of Total by AC | BG | % of Total by BG |
| Soil protection (SP) | 3,245,142 € | 99% | 3,548,257 € | 102% |
| Carbon sequestration (CS), market price | −304,189 € | (9%) | −342,031 € | (10%) |
| Support to extensive animal production (SEAP) | 139,423 € | 4% | 132,882 € | 4% |
| Plant food production (PFP) | 1301 € | 0% | 2707 € | 0% |
| Fiber production (FP) | 123,752 € | 6% | 124,585 € | 4% |
| Total | 3,205,429 € | 100% | 3,466,401 € | 100% |
| Carbon sequestration (CS), social cost | −215,228 € | (6%) | −242,003 € | (7%) |
| Carbon sequestration (CS), shadow price | −528,025 € | (17%) | −593,714 € | (18%) |

### 3.3. Extrapolation for the Regional Level

The TEV for each ecosystem is presented in Table 5. The TEV calculated for grasslands and *dehesa* lies between 146 €/ha/year and 176 €/ha. Here, the estimated values are mainly due to SP, followed by FP and then by SEAP. The TEV calculated for croplands lies between 41 € and 166 €. This wide range is due to the fact that the estimated ES values differ a lot across approaches, with the most significant difference occurring in SP. While in the class approach SP is valued at 73 €/ha, this value is 138% higher according to the buffer approach, which yielded 173 €/ha. For croplands in the region, PFP is almost irrelevant with both approaches (between 0.6 €/ha and 1.3 €/ha). At the TEV level, CS continues to present a negative contribution in both modelling approaches and ecosystems.

From national statistics, the area of permanent pastures in the region is 1,151,238 ha [13]. If this area corresponds to grasslands and *dehesa*, and the proportion of land uses verified on the dataset applies, there is an estimated area of 112,305 ha of croplands associated with PBBF in Alentejo. Taking these areas into consideration and applying them together Equation (5) with the TEV presented in Table 5, we arrive at the regional results presented in Figure 4, for

the class approach. The results for BG are obtained directly from each ES value multiplied by the total area considered for Alentejo.

**Table 5.** Total Ecosystem Value (in euros—€/ha/year) per considered ecosystem.

| Ecosystem Service | Approach by Average Class | | Approach by Buffer Generation | |
|---|---|---|---|---|
| | Grasslands and *Dehesa* | Croplands | Grasslands and *Dehesa* | Croplands |
| Soil protection (SP) | 141.3 € | 72.8 € | 173.4 € | 173.4 € |
| Carbon sequestration (CS), market price | −10.5 € | −32.8 € | −12.0 € | −13.3 € |
| Support to extensive animal production (SEAP) | 6.4 € | - | 6.0 € | - |
| Plant food production (PFP) | - | 0.6 € | - | 1.3 € |
| Fiber production (FP) | 8.6 € | 0.3 € | 8.7 € | 4.8 € |
| **Total ecosystem value** | **145.8 €** | **40.8 €** | **176.0 €** | **166.2 €** |
| Carbon sequestration (CS), social cost | −7.5 € | −23.2 € | −8.5 € | −9.4 € |
| Carbon sequestration (CS), shadow price | −18.3 € | −57.0 € | −20.8 € | −23.0 € |

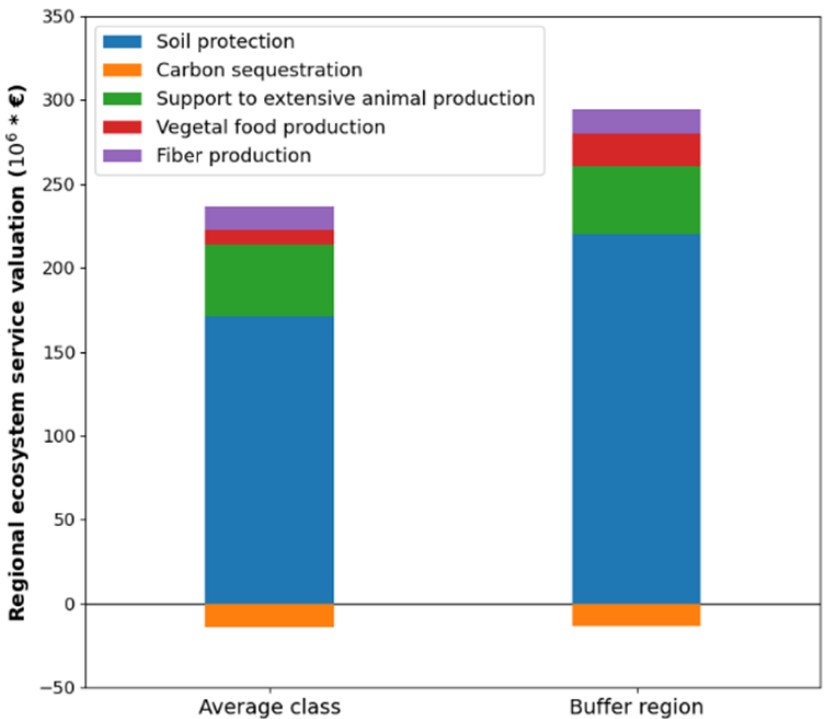

**Figure 4.** Estimated value of ecosystem services (€) from pasture-based beef farming in Alentejo after regional extrapolation for both modelling approaches.

Figure 4 depicts the total ES estimated value promoted by PBBF in Alentejo according to both modelling approaches. SP is the most valuable service, accounting for around 171 M€ and 220 M€ with class and buffer approaches, respectively. FP is the second most valuable ES, accounting for nearly 9.9 M€ in the first approach and 10.5 M€ in the second. It is followed by SEAP, that accounts for nearly 7.4 M€ in the class approach but just 6.9 M€ with the buffer approach. PFP presents a residual value of 68 K€ in the class approach and 154 K€ in the buffer approach. CS present similar negative values with the two methods, with values between −16 M€ and −15.5 M€. In total regional terms, ES provided by PBBF are estimated to be worth around 173 M€ using the class approach and around 223 M€ using the buffer approach (nearly 29% more than class approach).

## 4. Discussion

In both modelling approaches, SP is the ES that contributes most to the total value. This means that the main positive effect of animal production in Alentejo is soil protection, which is especially relevant, given that this is a region prone to desertification. SP is followed by FP, SEAP and then PFP. Concerning SP, land cover characteristics are, in general, very satisfactory in the territories where PBBF develop their activity, so the high service levels obtained are aligned with expectations based on knowledge and experience from the field. In the approach by class, SP presented significant differences between grasslands/*dehesa* ecosystems and cropland ecosystems (28.1 vs. 14.5 t/ha). The buffer approach considers the exact location of each farm and thus the local characteristics of the terrain in terms of soil and slope; accordingly, it returned a higher level of service of 34.5 t/ha of avoided erosion for both grasslands and croplands. These results suggest that in the studied sample of PBBF, both grasslands/*dehesa* and croplands provide overall higher levels of SP than grasslands/*dehesa* and croplands in general in the region.

The SEAP service is influenced by the average livestock density of cattle and sheep calculated at the municipality level, so it does not represent the real service level at the farm's location. A limitation of this indicator is that it does not take into account the proportion of livestock that is effectively in extensive production (or not). Nevertheless, in regional aggregated terms, it can be considered as a useful indicator, since the predominant animal production system in Alentejo is extensive. The values estimated for grasslands and *dehesas* are very similar across approaches, i.e., 0.4 and 0.3 LU per ha, both of which are below the regional average of 0.5 LU per ha, which suggests a lower livestock density in PBBF than in the remaining cattle and sheep production systems. The valuation applied for this ES is one of the main innovations of this work, since it intends to avoid double counting, a common mistake made in many similar studies. By using the rent values per ha (without subsidies) and associating it with the corresponding fraction of the animal production output, it is possible to ensure that the valuation refers exclusively to the added value provided by the land and its intrinsic characteristics. Other types of valuations usually fail to separate the added value coming from the different production factors as capital and labor, which can lead to double counting. There would also be another possible approach for this ES valuation, based on willingness to pay (WTP). In this case, there is a reference to an additional WTP for beef from pasture of 1.5 € per kg of meat [5]. However, that estimation was obtained in a study that measured WTP for a bundle of management practices. It is impossible to distinguish the fraction of that WTP that is specifically attributable to the support of extensive animal production, among the other benefits that beef from pasture may provide, which could lead to an overcounting problem.

Concerning PFP, the service levels obtained come from an aggregated regional basket of different crops. In terms of grasslands and *dehesa*, the PFP service level is zero, regardless of the approach. For croplands, the approach by class presented an overall service level of 1.5 t/ha, and the result through buffer generation was 3.3 t/ha. This evidences that croplands near the studied PBFF present a higher productivity than croplands in general in Alentejo. As many of the considered crops from the reference basket are not usually representative of PBBF, one improvement for future work would be to refine the crop basket to make it more representative of the most common crops associated with PBBF. The valuation technique applied here was the same as in SEAP, but in this case, it considered the fraction of vegetal production output.

For FP, the estimated service levels are 0.4 m$^3$/ha in grasslands and *dehesa* for both approaches. Using the class approach for croplands, there is a null service level (0 m$^3$/ha), while for the buffer approach, the same ecosystem has a service level of 0.2 m$^3$/ha. This suggests that croplands associated with PBBF territories present a higher service level. The FP service level was estimated directly through the mean annual increments of forests trees, deducting biomass losses due to natural mortality. The monetary value only considers the price of the main trees explored for timber, in regional terms. This means that all the other forest products were not considered in this indicator. In the specific case of the

Alentejo region, this could be a relevant limitation, since nearly 71% of the forest area is devoted to alternative productions, such as cork (very economically important), feed for autochthonous pigs, etc. Increasing the scope of the FP service in the future could be an interesting way for better accountancy of the ES provided by the forest fraction on PBBF. For now, this was impossible due to a lack of regionalized data on the other products from forests.

CS is the only ES that presents a negative contribution for the ES total value under both approaches. The CS for croplands was estimated at −0.6 tC/ha using the class approach and at −0.3 tC/with the buffer approach. For grasslands and *dehesa*, the obtained service levels are 0.2 tC/ha for both approaches, suggesting that the areas covered by these ecosystems emit small quantities of carbon instead of sequestering it, as could be expected. These results can be explained by the fact that the method for carbon sequestration calculations only considers land use change transitions as reported in the Portuguese NIR [19]. This means that any other aspects concerning the territory characteristics and/or specific practices that are applied in PBBF to promote carbon sequestration were not considered for this service level calculation. This limitation derives mainly from the high heterogeneity that characterizes PBBF and the territories where they develop their activity.

In this study, the average value of ES of grasslands was estimated as 146 €/ha/year (class approach) or 176 €/ha/year (buffer approach). Those are relatively low estimates compared with those in the relevant literature. For example, [28] estimated a global ES value for grasslands of 232 €/ha/year. Some other studies performed at a macro level estimated values of ES from grasslands of between 249 €/ha/year and 2352 €/ha/year [29], but in this case accounting for additional ES such as water treatment, supply and purification, gene pool protection (conservation) and cultural services. Finally, a study also based on PT-MAES data presented a value of 331 €/ha/year for permanent pastures in a natural park in Portugal [30]. Here, the different values are mostly explained by the different valuation methods applied.

Extrapolating those results to the entire region, the total estimated value for the considered ES provided by PBBF in Alentejo would amount to 173 or 223 M€, depending on the approach. The relevant difference between approaches suggests that the areas around the studied PBBF generate a higher overall ES value, when comparing with the same ecosystem areas in general. Here, it is necessary to point out that the total area considered for regional extrapolation does take in account pastures under tree cover, a relevant fraction of the area of the region [31]. Consideration of these areas would probably increase the overall ES levels and value.

According to collected data (and farmers' perspectives), there is no relevant PFP or FP in most of the surveyed farms, so the importance of these ES, as well as the relevance of SP, which is usually not perceived, can be considered another of the most surprising outcomes of this work. The absence of biodiversity indicators in this study is also notable, since many of the concerned areas present high nature value [32]. These limitations probably make the presented values fall short of the real value of ES provided by PBBF in Alentejo. The inclusion of biodiversity also likely made the valuation of ES in this study increase to levels like those cited in the literature. Nevertheless, comparing the scale of values obtained with the regional extrapolation—comprising nearly half of the national output of the beef cattle sector [13]—indicates the relevance of the value generated by the studied ES.

A potential future improvement in the continuation of this work would be to match the buffers generated to the correspondent real area of each farm. In fact, any in situ measurements that could be taken would improve the reliability of the results. From the farmers' perspective, it could be interesting to assess the main ES provided in each farm, especially if with that, the farmer could valorize his/her production and/or apply better management practices. When designing policies and incentives, the farm-size related particularities should also be considered to ensure equity and parity across the sector. It could also be argued that despite the generated ES, PBBF have significant environmental impacts that are also not valued by the market. This is the case particularly for beef

systems with methane and nitrous oxide emissions from enteric fermentation and manure management. A valuation of those emissions was beyond the scope of this work but should be carried out in the future and compared with the ES value obtained here.

## 5. Conclusions

The goal of this paper was to estimate the value of five ecosystem services provided by the areas covered by pasture-based beef farms in Alentejo. We obtained values for those ecosystem services of 146–176 €/ha/year for grasslands and *dehesa* (*montado*) and 41–166 €/ha/year for croplands. These results were robust to methodological choices, as we used two options for joining pixel-level ecosystem services valuations with farm-level production data. The two approaches provided similar results for the five studied services. Soil protection was the most valuable service, with the other studied services making significantly lower contributions. This led us to conclude that the main benefit of animal production for ecosystems in the region is the avoidance of soil loss. This result is particularly significant in Alentejo, which a highly desertified part of the country and of Europe. Regional extrapolation allowed us to estimate the overall value of ecosystem services associated with pasture-based beef farms in Alentejo: between 173 M€ and 223 M€, i.e., almost half of the national production value of the beef cattle sector. Thus, our results suggest that the maintenance of grasslands and croplands in pasture-based beef farms in Alentejo generates positive externalities for society. We therefore conclude that this production system is very important for the region in terms of the value generated by the studied ecosystem services. This work explored innovative ways of valuating ecosystem services, presenting two methodologies based on data available at the European level and a valuation method that avoids double counting. The present research is intended to address to the growing need for ecosystem services accounting in environmental and sustainability studies. Future studies should compare the positive contributions of the systems quantified here with the overall environmental impacts of the animal production systems in Alentejo. Starting with the conclusions of this study about the importance of pasture-based beef farms systems in the region, further work should also explore and engineer innovative and creative ways of maximizing the positive effects of those systems, as well as matching remuneration and incentives to these farmlands with the true value they generate for society.

**Author Contributions:** Conceptualization, M.P.d.S. and R.F.M.T.; methodology, M.P.d.S., T.G.M. and T.D.; formal analysis, data curation and writing—original draft preparation, M.P.d.S.; writing—review and editing, R.F.M.T., T.G.M. and T.D.; supervision, R.F.M.T. and T.D.; project administration, R.F.M.T.; funding acquisition, R.F.M.T. and T.D. All authors have read and agreed to the published version of the manuscript.

**Funding:** This work was supported by Fundação para a Ciência e Tecnologia through project "LEAnMeat—Lifecycle-based Environmental Assessment and impact reduction of Meat production with a novel multi-level tool" (PTDC/EAM-AMB/30809/2017), and by grants SFRH/BD/147158/2019 (dos Santos, M.P.) and CEECIND/00365/2018 (Teixeira, R.). The work was also supported by FCT/MCTES (PIDDAC) through project LARSyS—FCT Pluriannual funding 2020–2023 (UIDB/EEA/50009/2020).

**Data Availability Statement:** Not applicable.

**Conflicts of Interest:** The authors declare no conflict of interest. The funders had no role in the design of the study; in the collection, analyses, or interpretation of data; in the writing of the manuscript; or in the decision to publish the results.

# Appendix A

**Table A1.** Animal future Questionnaire items.

| General Information | Social Compartment | Economic Compartment | Animal Compartment |
|---|---|---|---|
| Own Land (ha) | Number Of Working Hours Rate | Total Output Vegetal | LivestockType (Cattle/Sheep) |
| Rented Land (ha) | Weekends Off | Total Output Animal | Categor (per age) |
| Legal Form (Individual/Company/Others) | Sundays Off | Total SpecificCosts | Average Number (No.) |
| Farmer Since (years of experience) | Days Off Holidays | Total Farming Overheads | Number Of Produced Animals (No.) |
| Family Labour (hours/year) | Workload Rate | Taxes | Age At End of Fattening (months) |
| Hired Labour (hours/year) | Hazardous Chemicals | Total Subsides On Crops | Number Of Sold Animals (No.) |
| Conventional or Organic | Physical Work | Total Subsides On Livestock | Live Weight At Sale (kg) |
| Area In Natura 2000 (ha) | Overwork Stress Rate | Total Support For RD | **Diet Compartment** |
| Area In Conservation Land No Natura 2000 (ha) | Activities Besides Farm | Decoupled Payments | Diet name (for all herd/fattenning/others) |
| Area Under Agro Environmental Measures (ha) | Work Life Balance Rate | Depreciation | Diet component name (ex: hay) |
| Area Of Specific Natural Habitat (ha) | Working Atmosphere Rate | Wages Paid | Diet Component Quantity (kgs/animal/year) |
| Overall Satisfaction being a Farmer (1–5) | Farm Economically Viable In 10 ys | Rent Paid | Protein Content Fraction (0–1) |
| **Other Information** | Expectation Farm Succession | Interests Paid | Considered number of animals (No.) |
| Outdoor Access Animals (y/n) | If Over 45 Succession Expected | Total Assets | Diet Component Quantity (kgs/farm/year) |
| Days Outdoor (0–365) | Training Days—Family Workers | Total Assets ExclLand | **Crop Areas Compartment** |
| Additional Enrichment (y/n) | Training Days—Staff | Liabilities | Surface (ha) |
| Animal Social Contact (y/n) | Highest Educational Degree | Own Labour Force Persons | Yield (t/ha) |
| Non Curative Treatments (y/n.) | Highest Agricultural Educational Degree | Own labour Force Hours | Fraction Of Legumes (0–1) |
| **Resources Utilization** | Public Access To Farm | Hired Labour Force Persons | Dry Matter Content (0–1) |
| Diesel Consumption (l) | Visits To Farm | Hired Labour Force Hours | **Crop Protection Agents** |
| Electricity Consumption (kw) | Professional Organisations Besides Farm | Imputed Labour Costs | Type (natural/artificial) |
| Renewable Energy Fraction (0–1) | What Professional Organisations Besides Farm | Rented Farm Land | Name |
| Irrigated Area (ha) | Non Professional Organisations Besides Farm | Rental For Farm Land | Quantity (m$^3$ or t/ha) |
| Water Use Animals (m$^3$) | Direct Selling or Tasting | Own Farm Land | **Fertilizers** |
| Water Use Irrigation (m$^3$) | Labelling Schemes | Interest Rate | Type (natural/artificial) |
| Water Use Total (m$^3$) | What Labelling Schemes | Profit | Total Quantity (m$^3$ or t/ha) |
| | Other Activities On Farm | Total Subsidies | N content (0–1) |
| | What Other Activities On Farm | Total hours | |

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
