# Peer review of "Valuing Ecosystem Services Provided by Pasture-Based Beef Farms in Alentejo, Portugal"

_land, doi:10.3390/land11122238_

Round 1

Reviewer 1 Report

Comments and Suggestions for Editor and Authors

The question of "Valuing ecosystem services provided by pasture-based beef farms" is a relevant and important topic.   It is very important for the academic community to develop research on these current issues, which greatly contribute to sustainability. The abstract is well written and well dimensioned. The introduction and literature review are written in a thoughtful and balanced way. The methodology and the discussion of the results are presented in a clear and objective way. The conclusions are presented in a clear and objective way. The bibliography is extensive and current.

Other issues related to the work presented:

1.   What is the main question addressed by the research? Is it relevant and interesting?

This paper aims to measure and value the ecosystem services of grasslands and croplands covered by pasture-based beef farms in Alentejo. This is a relevant, pertinent, and interesting question as a current research topic.

2.     How original is the topic? What does it add to the subject area compared with other published material?

 The topic is original and relevant in this area of investigation. The way the authors approach to measure and value the ecosystem services of grasslands and croplands covered by pasture-based beef farms in Alentejo is quite relevant, in terms of research and contribution to knowledge. This work was a pilot study within the framework of the European Union Biodiversity Strategy to 2020, a pan European initiative launched by the European Commission. This is an important work in this area of investigation.

3.     Is the paper well written? Is the text clear and easy to read?

 Yes, the text is well written, in a clear, objective, and precise way, but on Line 247 the text has incorrect words. Needs to be erased.

4.    Are the conclusions consistent with the evidence and arguments presented? Do they address the main question posed?
The conclusion is brief, and light. The conclusion must mention the objective of the work, its central question and its contribution to the advancement of knowledge. This part of the paper needs improvement.

5.     Is there any current bibliography that can be suggested, with the aim of improving the paper?

No, but the final bibliography must be standardized, according to the requirements of the Journal.

Author Response

We thank you Reviewer 1 for the comments, as they encouraged us to continue and improve our research work. Regarding the Reviewer’s two main remarks:

  1. (…) the text is well written, in a clear, objective, and precise way, but on Line 247 the text has incorrect words. Needs to be erased.

We corrected the mistake in line 247. This change also led us to correct the subtitle of Figure 2.

  1. “The conclusion is brief, and light. The conclusion must mention the objective of the work, its central question and its contribution to the advancement of knowledge. This part of the paper needs improvement.”

We agree that the conclusion was brief and missed important elements. We have now reformulated the Conclusions section. We now clearly state the objective of the work, its central question and its contributions to the advancement of knowledge.

We hope these changes fulfill the requests/suggestions of the Reviewer, and if necessary we are happy to continue improving the manuscript in future iterations.

Reviewer 2 Report

The presentation reflects the present state of knowledge. The paper is very well structured. The Introduction section is good, in this section the authors present clearly the objectives and the main contributions of the study. The authors provided sufficient background and include relevant references. The research design is innovative and appropriate. The method is adequately described. The results are clearly presented. The conclusions are supported by the results. 

Author Response

We thank Reviewer 2 for the positive and encouraging opinion of our work.